Genetic diversity and structure of the noble crayfish populations in the Balkan Peninsula revealed by mitochondrial and microsatellite DNA markers

Gross Riho 1
Lovrenčić Leona 2
Jelić Mišel mjelich@gmail.com misel.jelic@gmv.hr 3
Grandjean Frederic 4
Ðuretanović Simona 5
Simić Vladica 5
Burimski Oksana 1
Bonassin Lena 2
Groza Marius-Ioan 6
Maguire Ivana imaguire@biol.pmf.hr 2
1 Chair of Aquaculture, Estonian University of Life Sciences , Tartu , Estonia
2 Department of Biology, Faculty of Science, University of Zagreb , Zagreb , Croatia
3 Department of Natural Sciences, Varaždin City Museum , Varaždin , Croatia
4 Université de Poitiers , Poitiers , France
5 University of Kragujevac , Kragujevac , Serbia
6 University of Oradea , Oradea , Romania
Waiho Khor
Electronic publication date: 2021 Aug 4
Publication date: 2021
Volume: 9
Electronic Location ID: e11838
Received 2021 May 19; Accepted 2021 Jul 1
Copyright: ©2021 Gross et al.
Copyright year: 2021
Copyright holder: Gross et al.
License: This is an open access article distributed under the terms of the Creative Commons Attribution License, which permits unrestricted use, distribution, reproduction and adaptation in any medium and for any purpose provided that it is properly attributed. For attribution, the original author(s), title, publication source (PeerJ) and either DOI or URL of the article must be cited.
License URL: https://creativecommons.org/licenses/by/4.0/

Keywords: Astacus astacus, South-east Europe, Cytochrome oxidase subunit I, Conservation, Glacial refugia

Funding: Croatian Science Foundation CLINEinBIOta - IP-2016-06-2563 ESF - DOK-2018-01-9589 Estonian Ministry of Education and Research IUT8-2 Estonian Research Council PRG852 Serbian Ministry of Education, Science and Technological Development 451-03-9/2021-14/200122 This research was funded by the Croatian Science Foundation (CLINEinBIOta - IP-2016-06-2563 to Ivana Maguire, ESF - DOK-2018-01-9589 to Leona Lovrenčić), the Estonian Ministry of Education and Research (IUT8-2 to Riho Gross), the Estonian Research Council (PRG852 to Riho Gross) and the Serbian Ministry of Education, Science and Technological Development (Agreement No. 451-03-9/2021-14/200122 to Vladica Simić and Simona Duretanović). The funders had no role in study design, data collection and analysis, decision to publish, or preparation of the manuscript.

==============================
Background

The noble crayfish (Astacus astacus) is a native European species in decline, with a contracting range and diminishing populations and abundance. Previous studies revealed this species significant genetic diversity in the south-eastern Europe, with populations from the western and the southern part of the Balkan Peninsula being the most divergent. However, sampling of populations from the western part of the Balkans was limited and insufficient for investigating genetic diversity and population divergence for the purpose of conservation planning and management. Thus, the major aim of this study was to fill in this knowledge gap by studying mitochondrial and microsatellite DNA diversity, using 413 noble crayfish from 18 populations from waterbodies in the western part of the Balkan Peninsula.

Methods

Phylogenetic analysis of studied populations and their mitochondrial diversity were studied using COI and 16S sequences and population genetic structure was described using 15 microsatellite loci.

Results

Phylogeographic analysis revealed new divergent mitochondrial haplotypes for the populations in the westernmost part of the Balkan Peninsula in the tributaries of the Sava and Drava rivers. Microsatellite data indicated that these populations harbour an important component of genetic diversity within A. astacus. The results suggest that the western part of the Balkans played an important role as microrefugia during the Pleistocene climate fluctuations, allowing the long term persistence of A. astacus populations in this region. These results will also be important to supporting conservation decision making and planning.

Introduction

Freshwater crayfishes are important organisms for normal functioning of freshwater food webs in many parts of the world (Usio & Townsend, 2004) and they are considered flagship species for the conservation of aquatic systems (Füreder & Reynolds, 2003), hence an understanding of their biodiversity and conservation status is a priority (Souty-Grosset & Reynolds, 2009). One of the most widely distributed native European freshwater crayfish species is Astacus astacus (noble crayfish) whose range and abundance have declined rapidly due to a negative anthropogenic influence upon their habitats (e.g., fragmentation, destruction and pollution), overfishing and adverse impacts of invasive alien crayfish species (Kouba, Petrusek & Kozák, 2014). Besides being able to outcompete native crayfish (Hudina et al., 2016), invasive crayfish are also vectors of the pathogen Aphanomyces astaci, the causative agent of the crayfish plague, which is mostly lethal to native European crayfish species (Alderman, Holdich & Reeve, 1990; Kozubíková-Balzarová & Horká, 2015). As a result, A. astacus is listed by the IUCN as a Vulnerable species (Edsman et al., 2010). Also, it is listed in the Appendix III of the Bern Convention, and in Appendix V of Habitat Directive (92/43/EEC). In order to ensure that noble crayfish conservation is effective, it is necessary to develop management plans based on sound knowledge of the species ecology, biology and genetics (Schulz & Grandjean, 2005; Souty-Grosset & Reynolds, 2009).

Further complicating the understanding of the diversity within the noble crayfish is that it has been frequently translocated, especially in the central and northern Europe (Policar & Kozák, 2015; Jussila et al., 2016; Gross et al., 2017), as it is a valued human food source, potentially over millennia, and is economically important, attracting a premium price in the market place. As a consequence, the natural distribution of genetic variation in the noble crayfish is thought to have been impacted by historical introductions in many European regions (Schrimpf et al., 2011; Schrimpf et al., 2014; Gross et al., 2013; Gross et al., 2017), necessitating comprehensive geographic sampling, to distinguish original and remnant populations from recent translocations, and identify potentially missed or hybrid populations.

Nevertheless, it is presumed that noble crayfish populations in south-eastern Europe have maintained their original genetic structure, since this species has been of little commercial interest in the region and although diminishing, may reflect historical evolutionary and phylogenetic patterns (Simić et al., 2008; Maguire, Jelić & Klobučar, 2011; Pârvulescu & Zaharia, 2014; Slavevska-Stamenković et al., 2016; Ðuretanović et al., 2017). Conversely, crayfish from south-eastern Europe (Croatia, Slovenia and Bosnia and Herzegovina) have been used for restocking of freshwaters in the central Europe that were devastated by crayfish plague in the late 19th century (cf. Jussila et al., 2016; Schrimpf et al., 2014). Thus genetic signatures from crayfish native to this region may be present in central Europe, further highlighting the need for a comprehensive understanding of the genetic structure throughout the species range.

Large-scale studies of mitochondrial cytochrome oxidase subunit I (COI), and the 16S rRNA sequences by Schrimpf et al. (2014) indicated the existence of four mitochondrial DNA (mtDNA) lineages within the noble crayfish in Europe, with populations from the south-eastern Europe (Black Sea basin) having the highest genetic diversity in Europe. The populations from the western part of the Balkan Peninsula were the most divergent; however, the authors further more detailed analysis of the south-eastern European populations as their sampling was not exhaustive. Recently, Laggis et al. (2017) studied populations from the southernmost part of the south-eastern Europe (Greece) using the same mtDNA markers. Their research included extensive sampling of noble crayfish and the results revealed the existence of two new noble crayfish mtDNA lineages (they called groups) endemic to Greece, and showed that newly discovered lineages possessed the highest haplotype richness and genetic diversity found so far.

The major aim of this study was to fill in the gap in our understanding of the genetic diversity of the noble crayfish in south-eastern Europe by studying mitochondrial and microsatellite DNA variation in 18 previously unstudied populations from this region. The first aim was to examine mitochondrial variation using the COI barcoding gene and 16S rRNA in A. astacus populations from less sampled parts of the south-eastern Europe and compare diversity and genealogical relationships with previously studied European populations using these markers (Schrimpf et al., 2014; Laggis et al., 2017; Mrugała et al., 2017). The second aim of this study was to describe genetic structure and characteristics of 18 A. astacus populations from waterbodies in the western part of the Balkan Peninsula using a suite of 15 microsatellite loci developed by Gross et al. (2017) to assist conservation planning and the protection of priority populations of this species in this region. Results of the present study will help guide efficient and effective conservation plans and management strategies for protecting the genetic diversity and maintaining the adaptive potential of A. astacus populations on the regional level in south-eastern Europe.

Materials & Methods

Sample collection and DNA isolation

This study used more than 400 noble crayfish samples from 18 populations in the western part of the Balkans (Fig. 1, Table 1). The specimen collection was conducted in Croatia, Romania, Serbia and Slovenia with the approval of local authorities (Croatia UP/I-612-07/18-48/148; Romania 408/CJ/27.11.2018; Serbia 324-04-10/2021-04 and 03 No. 026-419/2; Slovenia 35601-1262 150/2006-6 and 35601-135/2010-9). Crayfish were collected by hand or with trapped baited traps (Westman, Pursiainen & Vilkman, 1978; Kozák et al., 2015). Crayfish samples from Albania were purchased from a fisherman on the Prespa Lake. The Prespa Lake and Ohrid Lake are approximately 10 km apart, and waters from the Prespa Lake feed Ohrid Lake through underground karst channels. Out of the Ohrid Lake springs the only outlet, the Black Drim River, which flows in a north direction into Albania and thus into the Adriatic Sea. A pereiopod from each specimen was removed and preserved in 96% ethanol. Wild caught crayfish were then released back into the water. This sampling method does not harm crayfish as these appendages regenerate after the next moulting. Total genomic DNA was extracted from the pereiopod muscle tissue with the Sigma GenElute Mammalian Genomic DNA Miniprep Kit (Sigma-Aldrich, USA) following the manufacturer’s protocol and stored at −20 °C.

Figure 1 Geographical location of the studied Astacus astacus populations.

Details about sampling sites are provided in Table 1. Map was prepared in QGIS 3.10 software (available at: https://qgis.org/en/site/) and finished in GIMP 2.10 (available at: https://www.gimp.org/). In order to distinguish red and green circles on the map, we included letters R for red, and G for green.

Mitochondrial DNA analyses

Mitochondrial 16S and COI gene fragments were amplified and sequenced with primers 16Sar/16Sbr (Simon et al., 1994) and LCO-1490/HCO-2198 (Folmer et al., 1994) allowing comparisons to be made with 16S and COI sequences from previous studies on this species (Schrimpf et al., 2014; Laggis et al., 2017; Mrugała et al., 2017) available from GenBank. Polymerase chain reactions (PCR) for COI were prepared in a total volume of 25 µl containing 10–50 ng/µl DNA template, 1.5 mM Promega Buffer, 0.04 U HotStart Polymerase, 0.15 mM of each dNTP, 0.7 mM MgCl2 and 0.4 µM of each primer. PCR conditions for COI were as follows: initial denaturation at 94 °C for 3 min, followed by 35 cycles of denaturation at 94 °C for 45 s, annealing at 48 °C for 60 s and extension at 72 °C for 60 s, and the final extension of 10 min at 72 °C. The final reaction mix in a total volume of 10 µL for 16S gene contained 0.05 U GoTaq G2 HotStart Polymerase, 1.5 mM GoTaq FlexiBuffer, 0.2 mM of each dNTP, 0.275 µM of each primer, and 10–50 ng/µl of DNA template. PCR conditions for 16S were as follows: initial denaturation at 95 °C for 3 min, followed by 40 cycles of denaturation at 95 °C for 1 min, annealing at 52 °C for 1 min and extension at 72 °C for 1 min, and the final extension of 5 min at 72 °C. The purification of PCR products was performed with EXOAnP Mix (20 U/µl of Exonuclease I (New England Biolabs), 5 U/µl of Antarctic phosphatase (New England Biolabs). The sequencing of purified PCR products was prepared by Macrogen, Inc. (Seoul, South Korea), with the same forward primers used for the gene amplifications.

Table 1 Information on sampling sites.

Information on sampling sites with abbreviation (Abbr.), habitat type, country, river basin and major tributary, sea basin (BS, Black Sea; AS, Adriatic Sea), coordinates, putative refugial area (WBA, western Balkans; WBS, western Black Sea; SBA, southern Balkans), year of sampling, sample size and origin (status) of studied Astacus astacus populations.

Site	Abbr.	Habitat	Country	Basin/Tributary	Sea basin	Coordinates	Refugial area	Year	Sample size	Status	
Kočevska	KOC	river	Slovenia	Danube/Sava	BS	45.573N, 14.797E	W BA	2015	10	native	
Bloke	BLO	lake	Slovenia	Danube/Sava	BS	45.786N, 14.516E	W BA	2015	28	native	
Jaruga	JAR	river	Croatia	Danube/Sava	BS	45.048N, 15.225E	W BA	2008	23	native	
Plitvice	PLI	lake	Croatia	Danube/Sava	BS	44.879N, 15.615E	W BA	2008	9	probably introduced	
Maksimir	MAK	lake	Croatia	Danube/Sava	BS	45.831N, 16.026E	W BA	2016	30	introduced	
Kačer	KAC	river	Serbia	Danube/Sava	BS	44.222N, 20.280E	W BA	2014	29	native	
Motičnjak	MOT	lake	Croatia	Danube/Drava	BS	46.305N, 16.386E	W BA	2016	32	probably introduced	
Totovec	TOT	lake	Croatia	Danube/Drava	BS	46.345N, 16.469E	W BA	2016	30	probably introduced	
Jankovac	JAN	stream	Croatia	Danube/Drava	BS	45.522N, 17.684E	W BA	2016	30	native	
Vuka	VUK	river	Croatia	Danube	BS	45.438N, 18.258E	W BA	2016	31	native	
Resnički	RES	stream	Serbia	Danube/Velika Morava	BS	44.090N, 20.937E	W BA	2014	30	native	
Korenica	KOR	lakea	Serbia	Danube/Velika Morava	BS	44.228N, 21.413E	W BA	2014	17	native	
Gazivode	GAZ	lakea	Serbia	Danube/Velika Morava	BS	42.942N, 20.648E	W BA	2014	26	native	
Grliško	GRL	lakea	Serbia	Danube/Timok	BS	43.812N, 22.232E	W BA	2014	13	native	
Somesul	SOM	river	Romania	Danube/Tisa	BS	46.712N, 23.338E	W BS	2016	22	native	
Petresti	PET	river	Romania	Danube/Tisa	BS	45.909N, 23.559E	W BS	2016	11	native	
Bezid	BEZ	lake	Romania	Danube/Tisa	BS	46.413N, 24.878E	W BS	2016	9	native	
Prespa	PRE	lake	Albania	L. Ohrid/Black Drim	AS	40.865N, 20.944E	S BA	2014	33	native	
Notes.

a reservoirs

Sequences were edited in SEQUENCHER v. 5.3 (Gene Codes Corporation, Ann Arbor, USA) and aligned using MAFFT v.7.187 (Katoh & Standley, 2013). The final alignment for the COI gene fragment was 655 bp long, while for 16S was 475 bp long. New 16S and COI sequences were submitted to GenBank and BOLD data bases, and their GenBank accession numbers are MW726211 –MW726336 and MW726338 –MW726635 for 16S and COI sequences, respectively (Table S1 in Supplements). Additionally, all available sequences of 16S and COI genes of A. astacus were downloaded from GenBank (COI sequences from Schrimpf et al. (2014) and Laggis et al. (2017) were 350 bp-long, while sequences from Mrugała et al. (2017) were 635 bp long, and 16S sequences were 475 bp long). GenBank accession numbers of the haplotypes obtained in the present study, as well as the ones from Schrimpf et al. (2014), Laggis et al. (2017) and Mrugała et al. (2017) are reported in Tables S1 and S2.

In order to reconstruct phylogenetic tree that will be comparable with trees obtained in previous studies (Schrimpf et al., 2014; Laggis et al., 2017) we concatenated COI and 16S sequences, and used only those samples for which sequences of both genes were available. This made concatenated data set (used in phylogenetic reconstruction) smaller compared to the COI data set that was used for other analyses. The 350 bp long COI sequences and 475 bp 16S sequences from the same individual were concatenated and collapsed to unique haplotypes using FaBox (Villesen, 2007). The full concatenated COI/16S data set included 83 haplotype combinations (Table S2) and the final alignment was 825 bp long including a single gap-containing position observed in 16S fragment. The optimal models of nucleotide evolution for each partition of the concatenated data set were selected under the Bayesian information criterion (BIC) using the jModelTest (Darriba et al., 2012). The selected model for 16S was HKY+I, while for COI was HKY+G. Phylogenetic tree was reconstructed using the concatenated haplotypes in BEAST v.2.5.2 (Bouckaert et al., 2019). Since the null hypothesis of equal evolutionary rate throughout the tree was rejected at a 5% significance level, we used a relaxed uncorrelated lognormal clock model and the arthropod substitution rate of 2.3% pairwise sequence divergence for COI (0.0115 substitutions/s/Ma/l) (Brower, 1994) along with an estimated molecular clock for the 16S. The tree prior was set as the birth-death and independent substitution models were assigned to each mtDNA gene. The Markov Chain Monte Carlo (MCMC) analysis run comprised 300,000,000 generations, sampled every 10,000 generations. In order to determine convergence, the Effective Sample Size (ESS) values were checked in Tracer (Rambaut et al., 2018). The best fit tree was produced using the Maximum clade credibility tree option in TreeAnnotator 2.5.2 after the 20% of the sampled trees was discarded as burn-in.

Additionally, the phylogenetic relationships were estimated using Bayesian analysis (BA) in MrBayes ver.3.2. (Huelsenbeck & Ronquist, 2001; Ronquist & Huelsenbeck, 2003) with priors set according to the suggested model for each partition (16S: HKY+I, COI: HKY+G). Two separate runs with four Metropolis-coupled Monte Carlo Markov chains (MMCM) were performed for 10,000,000 generations, and trees were sampled every 1,000 generations. After effective sample size (ESS values > 200) for each parameter was confirmed with Tracer, the first 25% of sampled trees were eliminated as burn-in, and a 50% majority-rule consensus tree was constructed, with nodal values representing the posterior probabilities. Sequences of Pontastacus leptodactylus (Acc. No. KX279350) were used as an outgroup.

The median-joining network (MJ) approach (Bandelt, Forster & Röhl, 1999) was used in order to establish non-hierarchical phylogeographic and phylogenetic relationships among samples. To that end, three data sets were prepared; two including only COI sequences, and one including concatenated (16S+COI) sequences. The COI sequence data set I comprised 655 bp sequences obtained in this study, while data set II comprised 350 bp long sequences published in Schrimpf et al. (2011); Schrimpf et al. (2014)) and Laggis et al. (2017), combined with sequences obtained in this study and the study by Mrugała et al. (2017). The sequences obtained in this study and the study by Mrugała et al. (2017) were trimmed to 350 bp in order to match size of data sets from Schrimpf et al. (2011); Schrimpf et al. (2014)) and Laggis et al. (2017). This approach enabled us to associate COI haplotypes obtained in the present study to the COI haplotypes obtained in their research and indirectly to the lineages sensu Schrimpf et al. (2014) and groups sensu Laggis et al. (2017). Since one of our aims was to position our new samples into previously constructed phylogenies (Schrimpf et al., 2014; Laggis et al., 2017), we adopted their naming of lineages or/and groups. The MJ networks were generated using the program PopART v.1.7 (Leigh & Bryant, 2015) with all parameters set to default values.

Genetic diversity indices (number of segregating sites, number of haplotypes, haplotype diversity, nucleotide diversity, average number of nucleotide differences) were calculated in program DNASP v.6 (Rozas et al., 2017).

Analysis of molecular variance (AMOVA) (Excoffier, Smouse & Quattro, 1992) was performed in Arlequin v.3.5 (Excoffier & Lischer, 2010) in order to estimate hierarchical distribution of genetic diversity of A. astacus. Populations were grouped on the basis of major tributaries of the Danube River (Drava, Sava, Vuka, Velika Morava, Timok, Tisa) and Prespa Lake. Standard AMOVA computations were performed with three hierarchical levels: among groups (river basins/tributaries), among populations within groups, and within populations. The variance components were tested statistically by non-parametric randomisation tests using 10,000 permutations. Genetic differentiation among populations and river catchments was analysed through estimation of pairwise values of ΦST.

Microsatellite DNA analyses

A total of 19 species-specific tetranucleotide repeat microsatellite loci were amplified following the protocols and procedures described by Gross et al. (2017) with the following modifications; initial screening of a few Balkan A. astacus populations revealed that they possess much higher variability and wider allele size range at many loci than the eastern European (Czech Republic and Estonia) populations used in the study of Gross et al. (2017). Therefore, we split the single 19-plex microsatellite panel into two multiplex (10-plex and 9-plex) panels (Table S3). Only 15 loci were used for data analysis, as it later became apparent that at four loci, the allele size ranges still overlapped in some populations (Aast4_26, Aast4_47, Aast4_10 and Aast4_30) (Table S3). The PCR reaction (10 μl) contained 1x Type-it Multiplex PCR Master Mix (QIAGEN, Germany), 200 to 400 nM of each primer (Table S3 ), and ca 5 ng/ μl of DNA template. Touchdown program was used for PCR amplification: initial activation of 5 min at 95 °C, followed by 20 cycles of 30 s at 95 °C, 90 s at 60 °C, 30 s at 72 °C, with the annealing temperature decreasing 0.5 °C per cycle, followed by 10 cycles of 30 s at 95 °C, 90 s at 50 °C, 30 s at 72 °C and a final extension for 30 min at 60 °C. The multiplex panels of microsatellite loci were genotyped on Applied Biosystems 3500 Genetic Analyser (Life Technologies, USA) using internal GeneScan 600 LIZ Size Standard v2.0 (Life Technologies, USA) and microsatellite genotypes were scored using GeneMapper v.5 software (Life Technologies, USA). Microsatellite genotypes were initially scored automatically and were double-checked manually by two experts. Subsample of 50 individuals (12% of total 413 analysed individuals) was genotyped twice to check the genotyping consistency.

MICRO-CHECKER 2.2.3 (Van Oosterhout et al., 2004) was used to assess the potential presence of genotyping errors due to scoring of stutter peaks, large allele dropouts and null alleles. FSTAT v. 2.9.3.2 programme package (Goudet, 2001) was used for calculating allele frequencies, FIS and pair-wise FST values (as estimated by Weir and Cockerham’s θ), for estimating expected (unbiased genetic diversity) and observed heterozygosities (HE, HO) and rarefied allelic richness (AR), and for testing the significance of differences in average values of AR, HE and HO among groups of populations (1,000 permutations, two-sided tests). The private allelic richness (APrRar) was estimated by rarefaction method using HP Rare v.Feb-2-2009 (Kalinowski, 2005) and multiplied by number of loci in order to make it comparable to the number of observed private alleles (APr). GENEPOP v. 3.3 (Raymond & Rousset, 1995a) was used to test genotypic distributions for conformance to Hardy-Weinberg (HW) expectations and to test the loci for genotypic disequilibria. All probability tests were based on the Markov chain method (Guo & Thompson, 1992; Raymond & Rousset, 1995b) using 1,000 de-memorization steps 100 batches and 1,000 iterations per batch. Sequential Bonferroni adjustments (Rice, 1989) were applied to correct for the effect of multiple tests. Analysis of molecular variance (AMOVA) incorporated in ARLEQUIN v. 3.5.1.2 (Excoffier & Lischer, 2010) was used to partition genetic variance hierarchically between population groups, between populations within groups and among individuals within the populations. Populations were grouped based on major tributaries of the Danube River (Drava, Sava, Tisa, Velika Morava, Timok and Vuka) and the Prespa Lake.

Principal Coordinates Analysis (PCoA), implemented in PAST v. 4.05 (Hammer, Harper & Ryan, 2001), was used to explore and to visualize DA genetic distance (Nei, Tajima & Tateno, 1983) matrix between populations in multidimensional space. The DA distances were calculated using POPULATIONS v1.2.3.1 software (Langella, 1999).

Long-term effective population size (Ne) was assessed from the microsatellite data using the coalescent-based approach implemented in Lamarc v.2.1 (Kuhner, 2006), considering a mutation rate of 5 × 10−4 and a mixed mutational model with 30% KAM as in Gouin et al. (2011). We ran three replicates of two initial chains retaining 10,000 genealogies sampled every 200 and discarding the first 1,000 as burn-in, followed by one final chain retaining 30,000 genealogies sampled every 200 and discarding the first 5,000 as burn-in.

Combined mtDNA and microsatellite analyses

In order to visualise the distribution of A. astacus genetic diversity across geographic space and connect genetic diversity patterns to geographic features we applied Alleles in Space (AIS) package (Miller, 2005), with software implementation of Mantel test and interpolation of genetic landscape shape (GLS). The Mantel test was used to test for correlations between genetic and geographic distances using three datasets: (a) COI data set including all sequences (350 bp long), (b) COI data set including only sequences obtained in this study (655 bp long), (c) genotypes of 15 microsatellite loci. Mantel tests were performed at the individual level using an analogue of Nei’s genetic distances (Nei, Tajima & Tateno, 1983) between pair of individuals. For the Mantel test and other analyses run in AIS package, significance was tested using 10,000 permutations.

GLS interpolation was performed using two datasets: (a) full COI data set including all sequences (350 bp long) aiming to get insight into diversity on the European level, and (b) genotypes from 15 microsatellite loci in order to get insight into local diversity on the Balkan Peninsula. First, sampling sites were connected through a network based on the Delaunay triangulation, in which the simple mismatch molecular distances between connected sampling sites were calculated based on the molecular data obtained from all individuals. Surface calculation was based on midpoints of edges derived from Delaunay triangulation. Surface heights were calculated based on residual genetic distances.

The values of molecular distance were set in the mid-points of each connection in the network using the Alleles in Space (AIS) software (Miller, 2005). The raw molecular distances were interpolated. The matrix of the ‘elevation’ values was imported into QGIS 3.10 software (available at: https://qgis.org/en/site/) to generate molecular divergence surface image using the inverse distance weighted (IDW) algorithm, plotted over a map of Europe.

Results

Phylogenetic relationships among populations using mitochondrial COI and 16S

The phylogenetic tree inferred by BEAST using concatenated data set indicated six mostly unsupported previously described genetic lineages sensu Schrimpf et al. (2014) and groups sensu Laggis et al. (2017) (Fig. 2A), with all newly obtained concatenated haplotypes (Hap49, Hap50, Hap51, Hap52, Hap53, Hap54) nested within them. Most of the lineages diversified during Pleistocene, within the period between 1.7 and 0.5 mya (Fig. 2A). Phylogenetic reconstruction using BA in MrBayes revealed unresolved relationships among lineages/groups and weak nodal support with numerous polytomies (Fig. S1).

Figure 2 Bayesian phylogram and Median joining network.

(A) Bayesian phylogram based on the concatenated COI and 16S haplotypes of the noble crayfish using BEAST. Values at nodes represent posterior probabilities >0.5. Phylogenetic clades are represented as in Schrimpf et al. (2014) (Lineages 1–4) and Laggis et al. (2017) (Group 1 and 2), and the position of new haplotypes is indicated by circle at the end of the branch. (B) Median joining (MJ) network of concatenated sequences (COI and 16S) of the noble crayfish. Numbers of mutational steps are given as hatch marks. The size of the circle is proportional to the frequencies of the haplotype, with black dots indicating extinct ancestral or unsampled haplotypes. Haplogroups are represented by different colour as in (A).

New haplotypes from Serbian populations (Hap49, Hap52 and Hap53) grouped with haplotypes from Kosovo, Montenegro and Germany (Fig. 2A). Haplotype 51 from Slovenia and haplotype 54 from Montenegro recovered within the same unsupported clade together with haplotypes from Croatia, Austria and Czechia (Fig. 2A). The haplotype representing specimens from the Lake Prespa in Albania (Hap50) was positioned within the clade encompassing haplotypes from Greece (Fig. 2A).

The median-joining (MJ) networks for concatenated data set (Fig. 2B), as well as for two COI data sets (Figs. 3 and 4), were mostly congruent and they depicted haplotype relatedness and distribution within A. astacus. The MJ network based on long COI sequences (655 bp) revealed the existence of 11 unique haplotypes, separated by different number of mutational steps (Fig. 3). The analysis of short-sequences data set (350 bp) produced MJ network containing 60 haplotypes, with 12 established in the present study, six of them (ssh1, ssh4, ssh5, ssh6, ssh10, ssh11) obtained for the first time (Fig. 4). The MJ network based on the concatenated data set encompassed 83 haplotypes, six of them (Hap49-Hap54) obtained for the first time (Fig. 2B).

Figure 3 Median-joining network of COI sequences (655 bp long) obtained from Astacus astacus populations.

Frequencies of haplotypes are proportional to the size of circles. The black dots indicate the median vectors, and numbers of base pair changes are indicated by hatch marks. Circles are coloured according to samples affiliation to the river tributary/ river basin/ country (RS-Serbia, RO-Romania, HR-Croatia, AL-Albania, SI-Slovenia). Haplotypes are labelled from Lshm1 to Lshm11 (Lshm –long sequence haplotypes (samples) used also for microsatellites analyses). The three letter codes indicate sampling locality (for details see Table 1) with numbers of analysed specimens in brackets. In order to distinguish red and green circles in the network, we included letters R for red, and G for green.

Figure 4 Median-joining network of 350 bp long Astacus astacus COI sequences.

The black dots indicate the median vectors, and numbers of base pair changes are indicated by hatch marks. Frequencies of haplotypes are proportional to the size of circles. Purple colour presents share of previously haplotyped sequences (Schrimpf et al., 2014; Laggis et al., 2017; Mrugała et al., 2017) and yellow colour presents share of sequences obtained in the present study (marked as ssh–short sequences haplotypes: ssh1 to ssh11) (for details see Table S1).

In the long-sequences MJ network, haplotype Lshm9 (detected in the Slovenian populations BLO and KOC (Sava River, tributary of the Danube River)) and haplotype Lshm10 (discovered in Croatia, in the TOT and MAK populations (Drava and Sava, tributaries of the Danube River, respectively)) were separated by five mutational steps, whereas 10 bp changes were observed between Lshm9 and Lshm3, and 11 between Lshm9 and Lshm6 (Fig. 3). In the short-sequences MJ those two haplotypes (ssh10 and ssh11, respectively) were separated by one and three bp changes, respectively, from the closest haplotype that was recorded in Austria and Czech Republic (Fig. 4). The same was observed from the concatenated MJ; Hap51 (COI haplotype ssh10) and Hap25 (COI haplotype Aas18) were separated by one base pair change (Fig. 2B). Moreover, when BEAST tree was reconstructed with concatenated sequences the same grouping was observed; Hap51 and Hap25 form well supported subclade within unsupported lineage 4 sensu Schrimpf et al. (2014) (Fig. 2A), while this subclade was less supported in the Bayesian phylogram (Fig. S1). The Lsmh2/ssh2 COI haplotype was recorded in Croatian populations from the Drava River system (MOT and JAN) as well as in the Romanian populations (BEZ and SOM) from the Tisa River system where also Lshm4/ssh5 was recorded (Figs. 3 and 4). The former (ssh2 haplotype) is widely distributed in Europe (Fig. 4). Interestingly, in the populations MOT and JAN another COI haplotype was recorded (Lshm6/ssh7, Figs. 3 and 4) indicating presence of crayfish belonging to two distinct lineages/groups in the Drava River system. In concatenated data set, in both MJ and BEAST tree, Hap42 (COI haplotype ssh7) was closely related to Hap41 (COI haplotype Aas26) and Hap54 (COI haplotype ssh6) distributed also in Germany and Montenegro, respectively (Fig. 2). Some of the A. astacus specimens from Serbian waterbodies had ssh4 (in concatenated data set Hap53) haplotype, while most of them possessed the Lshm3/ssh3 haplotype (Figs. 3 and 4). Both haplotypes were geographically restricted to the Danube River basin tributaries Velika Morava and Timok. In addition, in concatenated data set the later formed Hap26 that is positioned close to haplotypes distributed also in Romania and Kosovo (Fig. 2). The remaining specimens from Serbia were collected from the Sava River system (KAC) and they carried haplotype Lshm1/ssh1 (Figs. 3 and 4) that in concatenated data set forms Hap49, positioned closely to haplotype Hap28 from Croatia (Fig. 2). The Lshm7/ssh8 haplotype was established in the Vuka population, and in concatenated data set (Hap43) analyses is positioned closely to haplotypes recovered from the Save River in Croatia, but also haplotypes found in Germany (Figs. 2, 3 and 4). Finally, haplotype Lshm11/ssh12 was found in Croatian populations, and in concatenated data set it formed Hap28 that was positioned central to haplotypes from Danube’s tributaries in Croatia, Serbia and Romania, but also from the Rhine River system (Figs. 2, 3 and 4).

Summarised results of genetic diversity indices of 655 bp long COI sequences revealed that populations from the Danube’s tributaries Sava, Drava and Tisa rivers possessed higher number of haplotypes and nucleotide diversity compared to other river systems that were characterised by only one haplotype (Table 2).

Table 2 DNA polymorphism indices.

DNA polymorphism indices calculated using 655 bp-long COI sequences obtained from 18 Astacus astacus populations grouped according the river system: n, number of specimens; S, number of segregating sites; h, number of haplotypes; Hd, haplotype diversity; standard deviation in brackets, Pi, nucleotide diversity; standard deviation in brackets, k, average number of nucleotide differences, BS, Black Sea basin, AS, Adriatic Sea basin.

River system	Basin	n	S	h	Hd	Pi	k	
Sava	BS	64	19	4	0.750 (0.0003)	0.01274 (0.0004)	8.344	
Drava	BS	44	20	3	0.681 (0.0003)	0.01423 (0.0004)	9.323	
Vuka	BS	16	0	1	0	0	0	
Velika Morava	BS	31	0	1	0	0	0	
Timok	BS	12	0	1	0	0	0	
Tisa	BS	30	2	3	0.421 (0.087)	0.0007 (0.0002)	0.437	
Prespa Lake	AS	13	0	1	0	0	0	
Total		210	34	11	0.876 (0.008)	0.0145 (0.0005)	9.473	

Genetic differentiation of populations based on pairwise comparison of COI sequences revealed very high ΦST-values indicating genetically isolated populations with limited gene flow (Table S4). However, some of the population pairs’ ΦST-values were extremely low and not statistically significant. If populations are grouped according to the Danube River tributaries (Sava, Drava, Velika Morava, Vuka, Timok, Tisa) and the Prespa Lake, values of genetic differentiation varied from 0.193 (Drava-Sava) to 1.00 (Prespa Lake vs Vuka, Velika Morava and Timok), again demonstrating genetically different groups (data not shown).

Results of AMOVA conducted on 655 bp long COI sequences grouped according to specimen’s affiliation to major tributaries of the Danube River (Drava, Sava, Vuka, Velika Morava, Timok, Tisa) and Prespa Lake, revealed that most of the variance is contained among population within groups (61.05%, P < 0.001; Table 3), followed by variance among groups (31.1%), and a small amount of the variance was found within populations (8.56%).

Table 3 Analysis of molecular variance.

Analysis of molecular variance using mitochondrial COI sequences (655 bp long) and 15 microsatellite loci of Astacus astacus. For mtDNA data, populations were grouped based on their affiliation to major tributaries of the Danube River (Drava, Sava, Vuka, Velika Morava, Timok, Tisa) and Prespa Lake. For microsatellite DNA data, populations were grouped based on their affiliation to major tributaries of the Danube River (Drava, Sava, Tisa, Velika Morava, Timok and Vuka) and the Prespa Lake.

DNA marker	d.f.	Sum of squares	Variance components	Percentage of variation	
Mitochondrial data					
Among groups	6	530.89	1.66	31.10	
Among populations within groups	11	391.22	3.21	61.05	
Within populations	192	88.46	0.46	8.56	
Microsatellite data					
Among groups	6	1285.12	0.92	14.62	
Among populations within groups	11	1096.26	2.34	37.18	
Within populations	778	2360.32	3.03	48.21	
Notes.

df, Degrees of freedom

Microsatellites analyses

Genetic diversity

A total of 180 alleles were observed across the 15 microsatellite loci with an average of 12.0 alleles per locus, ranging from 6 alleles at Aast4_7 to 26 alleles at Aast4_17 (Table S3). The average observed heterozygosity of the studied loci was 0.407 and varied from 0.254 (Aast4_2) to 0.518 (Aast4_17) (Table S3 ).

All microsatellite loci in studied crayfish populations were in linkage equilibrium (data not shown). Only a single population (JAN from Croatia) displayed significant deviation from expected HW proportions after applying sequential Bonferroni correction (Table 4). MICROCHECKER software provided evidence for putative null alleles at 4 out of 15 microsatellite loci in four populations (one to two loci per population, data not shown). However, as only 4 out of 270 tests were significant (1.5%, less than the expected Type-I error level), we decided not to exclude any loci from further analysis.

Table 4 Genetic diversity parameters inferred from 15 microsatellite loci for 18 sites of the noble crayfish (see Table 1 for full names of populations).

Population	Basin/tributary	Sea basin	n	P	A	Ar	Apr	AprRar	He	Ho	FIS	PHWE	
KOC	Danube/Sava	BS	10	0.73	2.07	2.05	0	0.30	0.344	0.333	0.032	NS	
BLO	Danube/Sava	BS	28	1.00	3.40	2.90	2	1.65	0.465	0.464	0.003	NS	
JAR	Danube/Sava	BS	23	0.93	3.47	3.05	4	3.00	0.562	0.577	−0.027	NS	
PLI	Danube/Sava	BS	9	1.00	2.87	2.79	2	2.25	0.422	0.380	0.106	NS	
MAK	Danube/Sava	BS	30	0.93	2.93	2.39	8	6.15	0.355	0.350	0.013	NS	
KAC	Danube/Sava	BS	29	0.93	4.07	3.37	10	9.30	0.522	0.547	−0.049*	NS	
Aver. Sava				0.92	3.14	2.76	4.33	3.75	0.45	0.44	0.03		
MOT	Danube/Drava	BS	32	1.00	3.87	3.34	2	2.85	0.573	0.562	0.020	NS	
TOT	Danube/Drava	BS	30	1.00	3.27	3.09	1	1.65	0.577	0.557	0.036	NS	
JAN	Danube/Drava	BS	30	1.00	4.13	3.26	3	1.80	0.557	0.529	0.051	<0.05	
Aver. Drava				1.00	3.76	3.23	2.00	2.10	0.57	0.55	0.04		
VUK	Danube	BS	31	0.87	2.67	2.33	0	0.75	0.404	0.411	−0.017	NS	
RES	Danube/V. Morava	BS	30	0.93	3.67	2.96	8	4.95	0.510	0.529	−0.037	NS	
KOR	Danube/V. Morava	BS	17	0.73	2.00	1.91	0	0.60	0.306	0.267	0.133	NS	
GAZ	Danube/V. Morava	BS	26	0.47	1.53	1.48	0	0.15	0.126	0.133	−0.063	NS	
Aver. V. M.				0.71	2.40	2.12	2.67	1.95	0.31	0.31	0.01		
GRL	Danube/Timok	BS	13	0.33	1.33	1.31	0	0.65	0.106	0.113	−0.069	NS	
SOM	Danube/Tisa	BS	22	0.93	2.53	2.22	1	0.30	0.334	0.342	−0.024	NS	
PET	Danube/Tisa	BS	11	0.73	1.93	1.89	1	1.05	0.313	0.309	0.013	NS	
BEZ	Danube/Tisa	BS	9	0.47	1.73	1.69	0	0.00	0.144	0.163	−0.143*	NS	
Aver. Tisa				0.71	2.06	1.93	0.67	0.45	0.26	0.27	−0.01		
PRE	L. Ohrid/Black Drim	AS	33	0.73	2.27	1.83	11	11.25	0.237	0.230	0.027	NS	
Notes.

BS Black Sea

AS Adriatic Sea

n sample size

P proportion of polymorphic loci

A average number of alleles/locus

AR mean allelic richness

Apr number of private alleles

AprRar rarefied values of number of private alleles

He expected and Ho, observed heterozygosity

FIS inbreeding coefficient

PHWE probability of deviations from expected Hardy–Weinberg proportions after sequential Bonferroni adjustments (15 simultaneous tests per population)

* P < 0.05.

Genetic variation, expressed as the proportion of polymorphic loci (P), mean allelic richness (AR) and observed heterozygosity (HO), was on an average higher in the populations from the Black Sea basin than in the Adriatic Sea basin (P = 0.82 and 0.73, AR = 2.5 and 1.8, HO = 0.386 and 0.230, respectively). Among the major tributaries of the Danube River, the highest average variation was observed in the Drava River populations (P = 1.00, AR = 3.2, HO = 0.549), while the lowest average variation was recorded in the Tisa River populations (P = 0.71, AR = 1.9, HO = 0.271) (Table 4). Overall, the most variable populations were JAR and KAC from the Sava River, MOT, TOT and JAN from the Drava River and RES from the Velika Morava River, while the least variable populations were GAZ from the Velika Morava River, GRL from the Timok River and BEZ from the Tisa River (Table 4).

The results of the hierarchical gene diversity analysis by AMOVA revealed that for the total data set, the highest percentage of variation was present within populations (48.21%), followed by variation among populations within river systems (37.18%) and among river systems (14.62%) (Table 3).

Of the total 180 alleles, 127 were shared by the Black Sea and Adriatic Sea basin populations, 11 alleles were confined to the Adriatic Sea basin and 42 alleles were found only in the Danube River basin of the Black Sea basin. Among major tributaries of the Danube, 26, 6, 8 and 2 alleles were confined to the Sava, Drava, Velika Morava and Tisa rivers, respectively (averaged values are shown in Table 4). When the number of private alleles was rarefied, the highest private allelic richness (average number of rarefied private alleles) was again confined to the Prespa population from the Adriatic Sea basin, followed by Sava, Drava and Velika Morava populations of the Danube basin, while the lowest private allelic richness was indicated for the Vuka, Timok and Tisa populations of the Danube basin (Table 4).

Genetic differentiation and population structure

The overall level of genetic differentiation between all studied samples was high (global FST = 0.501) with pairwise estimates of FST ranging from 0.211 (between JAR and MOT populations in Croatia) to 0.838 (between Serbian GRL and Romanian BEZ populations) (Table S4). The average level of differentiation among populations from different major tributaries of the Danube (Sava, Drava, Velika Morava, Tisa) was relatively high (average pair-wise FST ranging from 0.360 between Sava and Drava to 0.623 between Velika Morava and Tisa) but lower than their differentiation from the PRE population of the Adriatic Sea basin (average pair-wise FST from 0.578 between Drava and PRE to 0.736 between Tisa and PRE). Within the major tributaries of the Danube, populations were more differentiated in Sava, Velika Morava and Tisa (average pair-wise FST 0.425, 0.473 and 0.502, respectively) than in Drava (average pair-wise FST = 0.281) (data not shown).

PCoA analysis provided good resolution of spatial population relationships reflecting their affiliation to the river systems (Fig. 5). The Adriatic Sea basin PRE population was clearly separated from the Black Sea basin populations on both PCo1 and PCo2. Further, PCo1 clearly separated populations belonging to the Tisa, Timok and Velika Morava tributaries from the populations belonging to the Sava and Vuka rivers (Fig. 5). PCo2 separated populations from the Tisa River tributaries and populations from Serbia (both Timok and Velika Morava tributaries) (Fig. 5).

Figure 5 PCoA scatterplot of the first two coordinates.

See Table 1 for full names of populations. In order to distinguish red and green circles on the map scatterplot, we included letters R for red, and G for green.

Effective population size

Effective population size estimates varied strongly between populations and within river basins, ranging from 9 to 4920 (Table 5). The highest values were found in the Croatian populations of MOT, TOT and JAN located in the north-western part of the study area in the Drava River basin, with Ne values of 4920, 4864 and 4742, respectively. Globally, the Croatian populations displayed the highest effective population sizes (mean Ne = 2951), while lower estimates appeared mainly in the eastern populations of Serbia (mean Ne = 1113; the value drops to 539 if the north westernmost population of KAC from the Sava River basin is not considered), Romania (mean Ne = 272) and Albania (Ne = 491). The lowest Ne values were found in the Romanian population of BEZ (Ne = 11), and the Serbian populations of GAZ (Ne = 18) and GRL (N e = 9).

Table 5 Effective population size estimates - Theta (Θ), effective population size (Ne) and its 90% confidence interval (CI) estimated from the microsatellite data with Lamarc for the 18 Astacus astacus populations from the Balkans. Θ = 4Neµ.

µ= 5 × 10−4.

Pop.	Θ	Ne	90% CI	
KOC	0.668	334	200–572	
BLO	2.043	1021	962–1704	
JAR	4.485	2243	1987–3384	
PLI	1.117	558	336–790	
MAK	3.152	1576	1044–2106	
KAC	6.812	3406	2356–4800	
MOT	9.839	4920	4050–5041	
TOT	9.727	4864	3777–5005	
JAN	9.483	4742	3435–5006	
VUK	3.512	1756	1171–2224	
RES	3.628	1814	1288–2260	
KOR	0.634	317	288–560	
GAZ	0.037	18	14–38	
GRL	0.019	9	1–11	
SOM	1.193	597	319–643	
PET	0.419	209	166–345	
BEZ	0.023	11	8–17	
PRE	0.981	491	352–985	

Spatial analysis

Mantel’s tests showed significant correlations between genetic and geographical distances (r = 0.317 (PMantel < 0.001) for 655 bp long COI sequences dataset; r = 0.096 (PMantel < 0.001) for 350 bp long COI sequences dataset; r = 0.493 (PMantel < 0.001) for microsatellite dataset). Alleles in Space (AIS) analysis was used to visualise the general genetic divergence on the European level for the COI data set (Fig. 6). Also, AIS was used to delineate genetic divergence in the area of interest (Balkan Peninsula) for the microsatellite data set (Fig. 7). The GLS interpolation on the COI data set showed that the area of the highest genetic divergences for A. astacus was located in the southern part of the Balkan Peninsula (Greece, Albania and North Macedonia) (Fig. 6). Moderate level of genetic divergence was observed among populations in Croatia and Slovenia, whereas the lowest divergences were indicated among populations from southern Serbia, Romania, Germany and Poland. The GLS interpolation on the microsatellite dataset indicated the highest genetic divergences was across a large area in the western part of Balkans (covering Sava and Drava River basins in Croatia and Slovenia) and the area in the southern Balkans (approximately corresponding to North Macedonia) (Fig. 7). Areas with the lowest genetic divergence were from parts of Romania, Serbia and Hungary.

Figure 6 The genetic landscape map inferred by 350 bp long Astacus astacus COI sequences.

A genetic landscape map based on COI was overlaid onto a relief map of Europe. Black dots refer to the sampling sites, red colour presents high molecular divergence between neighbouring populations, while blue colour correspond to areas of lower molecular divergence among populations.

Figure 7 The genetic landscape map inferred by multilocus microsatellite genotypes of 18 Astacus astacus populations.

To avoid extrapolating beyond the spatial extent of collection points, the genetic landscape is clipped to the extent of the original network (sampling extent) and to the boundaries of the region of analysis. Details about sampling sites (abbreviations) are provided in Table 1.

Discussion

In the present study we have updated the current knowledge on the mtDNA diversity of A. astacus in Europe by analysing numerous unstudied populations in the western part of the Balkan Peninsula. Study revealed new haplotypes (both COI and 16S) that nested among haplotypes belonging to different lineages described in previous studies (Schrimpf et al., 2014; Laggis et al., 2017). Moreover, genetic structure of studied A. astacus population revealed by microsatellites indicated that populations in the western part of the Balkans harbour important components of genetic diversity for the species as anticipated in the previous studies (Schrimpf et al., 2011; Schrimpf et al., 2014; Schrimpf et al., 2017; Gross et al., 2013; Laggis et al., 2017). The finding of population structuring at both local, and larger geographic scales in this study is consistent with other studies of A. astacus across its distribution range (Schrimpf et al., 2011; Schrimpf et al., 2014; Schrimpf et al., 2017; Gross et al., 2013; Makkonen, Kokko & Jussila, 2015; Bláha et al., 2016; Laggis et al., 2017; Mrugała et al., 2017; Panicz et al., 2019).

Astacus astacus is characterised by the complex evolutionary history as it has a large distributional range across a number of large catchments, and phylogeographic patterns and genetic diversity shaped through past geo-climatic processes and recent anthropogenic activities (translocations, reintroductions) (Kouba, Petrusek & Kozák, 2014; Policar & Kozák, 2015). Climate oscillations during the Pleistocene followed by postglacial (re)colonization processes from southern refugia, shaped the biodiversity of current European fauna (Hewitt, 1999), including A. astacus. Populations isolated in the southern refugia and micro-refugia accumulated genetic variation throughout the glacial period, however some of that diversity was lost due to bottlenecks and the founder effects experienced by populations during subsequent range expansions to the north (Hewitt, 1999). The observed pattern of reduced genetic diversity indicated by low haplotype diversity, allelic richness and fewer private alleles, in the north and central Europe (areas with pronounced glaciations) point to more recent range expansion into these regions (Gross et al., 2013; Klobučar et al., 2013; Schrimpf et al., 2014; Laggis et al., 2017; Berger et al., 2018, this study).

Previous studies suggested that A. astacus populations persisted through glaciations on the Balkan Peninsula, in three refugia: (I) western part of the Balkan Peninsula (waterbodies of the Adriatic and the eastern Black Sea basins in Croatia and Montenegro (Schrimpf et al., 2014); (II) the eastern Black Sea basin (waterbodies of the lower Danube in Romania, Bulgaria, Hungary (Schrimpf et al., 2014); (III) southern Balkans (Greece) (Laggis et al., 2017). These suggestions are supported by our finding of high genetic diversity between and within populations on the Balkan Peninsula compared to those in central and northern Europe (Schrimpf et al., 2014; Laggis et al., 2017; Mrugała et al., 2017; this study). The most likely postglacial colonization route of A. astacus towards north and central Europe was through the Danube River system (Schrimpf et al., 2014; Laggis et al., 2017). Also, human activities, such as translocations and re/introductions of A. astacus, also probably strongly influenced the natural genetic structure and diversity (e.g., population mixing and introgression between introduced and indigenous populations) across Europe (Souty-Grosset & Reynolds, 2009; Schrimpf et al., 2011; Schrimpf et al., 2014; Gross et al., 2013; Gross et al., 2017; Makkonen, Kokko & Jussila, 2015).

Phylogenetic analyses based on the concatenated mtDNA data positioned newly obtained haplotypes within previously described haplogroups (Schrimpf et al., 2014; Laggis et al., 2017) which are mostly weakly supported. The median-joining network for concatenated data set as well as for 350 bp COI data set reflected those relationships, while the analysis of longer COI sequences (655 bp) refined relations and indicated existence of undescribed diversity. These results uncovered how length of COI sequences could influence on discrimination and relationships between haplotypes. Even though short barcode sequences are suitable for species identification, frequently they are not accurate for resolving phylogenetic relationships (Min & Hickey, 2007; Vecchioni et al., 2017; Meiklejohn, Damaso & Robertson, 2019) and therefore in the future studies that will use longer sequences data sets possibly clearer insight into phylogenetic relationships among different haplotype groups within A. astacus will be gained.

It is worth mentioning that the number of mutational steps between A. astacus mtDNA lineages/groups is much lower when compared to the number of mutational steps between mtDNA lineages within other European freshwater crayfish species, such as Austropotamobius pallipes (varying from 1 to 26) and Austropotamobius torrentium (varying from 1 to 36) (Fratini et al., 2005; Klobučar et al., 2013; Jelić et al., 2016). Compared to other European freshwater crayfish species, A. astacus exhibits lower genetic diversity (Klobučar et al., 2013; Maguire et al., 2014; Akhan et al., 2014; Jelić et al., 2016). Based on the Barcode of Life Data Systems (BOLD system; http://www.boldsystems.org) records, all published A. astacus COI sequences form a single BIN (barcode index number) (cluster), while other European crayfish species form 17, 5 and 5 BINs (A. torrentium, A. pallipes and Pontastacus leptodactylus, respectively). Therefore, it can be inferred that all A. astacus mtDNA lineages/groups that have been described up to now (Schrimpf et al., 2014; Laggis et al., 2017) belong to a single species with indication that southernmost populations (haplotypes from well supported Group 1 and 2 sensu Laggis et al. (2017) could present subspecies A. astacus balcanicus (Laggis et al., 2017)/ A. balcanicus balcanicus (Crandall & De Grave, 2017). Future studies will probably clarify/resolve this taxon status as suggested in the study of Astacus colhicus by Bláha et al. (2021). Nevertheless A. astacus displays relatively low level of genetic variation across its large geographic range compared to other members of the Astacidae family.

It is important to highlight the distribution range of haplotypes Lshm9 and Lshm10, that is limited to the westernmost part of the Drava and Sava tributaries in Croatia, and in the Sava River tributaries in Slovenia, likely indicating that this area of species distribution might have had an important role as microrefugia allowing A. astacus populations to survive Pleistocene climate fluctuations.

The present study established presence of crayfish with COI haplotype Lshm2 (ssh2, identical to COI haplotype Aas01 from Schrimpf et al. (2014)) in Croatia. This haplotype was recorded exclusively in the Drava River drainage, and is the most widely distributed haplotype in Europe. Moreover, haplotype Lshm2, as well as closely related haplotypes (Lshm4 and Lshm5) were recorded in the A. astacus populations from Romanian waterbodies. According to our results and Schrimpf et al. (2014), it can be presumed that this haplotype did not originate in the Croatian freshwaters, but rather in Romania. Presence of closely related haplotypes in tributaries of the Tisa and Drava Rivers provide evidence of historical hydrological connection between those rivers (both tributaries of the Danube River). Moreover, the microsatellites based genetic distances (observed in PCoA plot) also indicated closer relationships of Drava’s populations JAN and TOT to Romanian Tisza populations than to Sava or V. Morava populations.

Our results regarding populations TOT and MOT showed discrepancy between observations and expected results. Since the geographical distance between these two lakes is only 8 km, it was expected that specimens from both populations would share similar/identical haplotypes. However, this was not the case; the crayfish from TOT possessed distinct COI haplotype that grouped together with samples from MAK, BLO and KOC, while in the MOT population we discovered the presence of two distant haplotypes (Lshm2 and Lshm6). The observed genetic structure indicated complex evolutionary history of A. astacus in the Drava drainage that possibly played an important corridor for crayfish post-glacial range expansion. Moreover, it should be pointed out that both TOT and MOT are gravel pits that were part of the Drava River, so it is possible that these A. astacus populations represent remnant astacofauna formerly present in local river systems but now lost because of invasive species (Hudina et al., 2009) and therefore represent important ark sites (Peay, 2009). On the other hand, as the gravel pits are regularly used by fisherman the possibility of introduction of translocated crayfish from unknown locations cannot be excluded (Maguire, Jelić & Klobučar, 2011). Similar scenarios of crayfish translocations were found in different crayfish species across the globe: e.g., Australia (Nguyen et al., 2002) and Europe (Petrusek et al., 2017).

The Serbian noble crayfish populations possess widely distributed haplotypes that have pan-European distribution, without an apparent geographical pattern.

The Prespa Lake samples analysed in the present study, and the Ohrid Lake samples studied in Mrugała et al. (2017) share a single COI haplotype what possibly indicates a historical bottleneck that reduced their diversity. Possible bottleneck for PRE population was also indicated by relatively small effective population size, what is similar to findings in Laggis et al. (2017) for A. astacus in Greece, and Gouin, Grandjean & Souty-Grosset (2006); Gouin et al. (2011)) for A. pallipes in France.

The AMOVA results are not congruent between COI dataset and microsatellites what might be a consequence of different effective population sizes or/and mutation rates of the bi-parentally and maternally inherited markers (Chesser & Baker, 1996). While COI suggested that majority of variance exist among populations within major tributaries of the Danube River (Drava, Sava, Vuka, Velika Morava, Timok, Tisa) and Prespa Lake, AMOVA for microsatellite revealed that most of genetic variation is within populations, which is similar to findings in Schrimpf et al. (2017) and Panicz et al. (2019). Observed genetic structuring might be an indication that A. astacus used to be widely distributed, and nowadays populations are isolated within drainages (tributaries/basins) but still retain a part of their original diversity, which is also consistent with high ΦST pairwise values.

Genetic structuring of A. astacus populations in the studied area was influenced by isolation by distance at a moderate level as observed using both COI and microsatellites datasets. However, isolation by distance probably was not the only factor that contributed to the observed genetic differences between populations; significant differences could be a consequence of landscape characteristics that produced geographically isolated population which have no surface water connection. Hydrogeography and complex landscape character of studied area played an import role onto the genetic differentiation of different freshwater taxa (e.g., Previšić et al., 2009; Klobučar et al., 2013; Jelić et al., 2016). Furthermore, genetic structure of A. astacus populations in Europe was shaped through anthropogenic influence (Schrimpf et al., 2014; Laggis et al., 2017), and possibly unexpected similarity between KAC (Serbia) and JAR and PLI (Croatia) populations can be explained by artificial stockings between the two countries.

The COI data set GLS interpolation indicated that, on the European level, the entire Balkan area was an important refugium for A. astacus in the past. Within the Balkan area, as shown by microsatellite GLS interpolation, there is a spatial subdivision of genetic diversity pattern within this large refugium suggesting several microrefugia (e.g., western and southern parts of the Balkan Peninsula) also known for the occurrence of distinct freshwater taxa, such as amphipods, fish or insects (Economidis & Banarescu, 1991; Previšić et al., 2009; Grabowski et al., 2017; Vucić et al., 2018).

Using microsatellite markers, we revealed high genetic diversity (12.0 alleles per locus) and high differentiation (FST = 0.512) among populations, however relatively low diversity within populations (on an average 2.8 alleles per locus), indicating a long-term isolation of small refugial populations. This strong population subdivision may limit local adaptation and facilitate random genetic drift which might result in diminished evolutionary potential of A. astacus (Nguyen et al., 2004; Steeves, Johnson & Hale, 2017; Hoffmann, Miller & Weeks, 2020). The pairwise FST values obtained in our analyses indicated clear signs of high differentiation among populations within different tributaries of the Danube River and the Prespa Lake, as well as between population-pairs. This study revealed the highest value of global FST compared to other studies (FST = 0.264 in Gross et al. (2013); FST = 0.232 in Schrimpf et al. (2014); FST = 0.400 in Laggis et al. (2017)).

Even though the number of samples per population was not even, application of microsatellite data rarefaction showed that results of rarefied and not rarefied data are congruent, and different sample number did not influence much the results. The highest genetic diversity, revealed by microsatellites, was found in JAR and KAC (Sava), MOT, TOT and JAN (Drava), and in RES (Velika Morava) populations. These crayfish populations exhibit the highest values of average number of alleles per locus, allelic richness, expected and observed heterozygosity of all studied populations. Furthermore, those genetic diversity estimates were higher than the ones reported in previous studies where the same set of tetranucleotide microsatellite loci were used (Gross et al., 2017). When comparing other microsatellite studies on A. astacus, higher values of genetic diversity estimates were detected in populations from the southern Balkans (Laggis et al., 2017), while lower values of genetic diversity estimates were observed in the populations from central, northern and eastern Europe (Gross et al., 2013; Schrimpf et al., 2014; Panicz et al., 2019). The values of indices obtained in the present study suggested the postglacial re-colonisation towards north followed by decreasing genetic diversity, as it is observed for numerous taxa (Taberlet et al., 1998; Hewitt, 1999). Since JAR, KAC, MOT, TOT, JAN and RES populations possess the highest reservoir of genetic diversity, and high effective population sizes, they could play an important role in future conservation programs. Crayfish from those populations could be a source for repopulation/restocking but bearing in minds their genetic background as well as composition of recipient population need to be taken into account in order to avoid inbreeding/outbreeding depression (Souty-Grosset & Reynolds, 2009; Hoffmann, Miller & Weeks, 2020). Furthermore, special effort should be taken in conservation of these populations not only because they harbour the highest genetic diversity, but also have the greatest number of private alleles, along with populations from PRE and MAK. Rare alleles are often considered a minor element in genetic conservation programmes, but they can be very important for the long-term response to selection and the survival of populations and species (Allendorf & Luikart, 2007). Moreover, PCoA analysis also revealed that the most distinct population was PRE (Black Drim/Adriatic Sea basin) and the rest of populations were well distinguished and grouped mainly according to their affiliation to river system. On the other hand, the lowest genetic diversity was recorded in GAZ (Velika Morava), GRL (Timok) and BEZ (Tisa) populations, which are also characterised with low effective populations sizes. These values of genetic diversity estimates are among the lowest compared to all previous population genetic studies (Gross et al., 2017).

In several populations the expected heterozygosity was higher than the observed heterozygosity indicating homozygote excess. In order to estimate the deviation from the Hardy-Weinberg equilibrium caused by inbreeding, which reduces the amount of genetic diversity in a population, we calculated the inbreeding coefficient (FIS; proportional to the loss of genetic diversity and consequently loss of adaptive evolutionary potential of the species (Frankham, 2005)). Since the FIS values in populations were slightly higher/lower than zero, we could conclude that inbreeding/outbreeding occurs, but is still not significant, and the intra-population variability is still evident.

Compared to previous population genetic studies of A. astacus (Gross et al., 2013; Schrimpf et al., 2014; Mrugała et al., 2017; Panicz et al., 2019), the overall genetic diversity in the study area was notably high. Our study confirms higher haplotype diversity and number of private haplotypes/alleles in the Black and Adriatic Sea basins compared to the North and Baltic Sea basins further corroborate a glacial refugium in the Balkan area.

It is necessary to know the genetic structure of the species in order to be able to preserve its integrity and within species diversity (Schrimpf et al., 2014). The combination of phylogenetic information and the degree of threat to species are both important for establishing conservation priorities (Owen et al., 2015). The results we obtained could be used as a starting point for developing future management plans. We would suggest that crayfish from populations with distant mtDNA haplotypes and, to some extent, between different tributaries of the Danube River (Sava, Drava, Tisa, Velika Morava, Timok) and the Prespa Lake should be treated separately in the future conservation projects (e.g., restocking/repopulations) that will require careful and balanced approach in order to avoid outbreeding and inbreeding depression (Schrimpf et al., 2017). Preserving genetic variability between and within A. astacus populations will ensure their evolutionary potential and long-time survival.

Conclusions

In the present study we updated the current knowledge on the mtDNA diversity of A. astacus in Europe by including previously understudied populations and geographic regions. New haplotypes were discovered restricted to the western part of the Balkan Peninsula. Analyses of microsatellites revealed population structuring at both local, and larger geographic scales observed across the A. astacus distributional range and indicates a complex genetic structure and confirmed that populations in the western part of the Balkans harbour important component of genetic diversity of the species. This information will help inform future conservation and management programs.

Supplemental Information

Supplemental Information 1 Supplemental Tables

Click here for additional data file.

Supplemental Information 2 Bayesian phylogram based on the concatenated COI and 16S haplotypes of the noble crayfish using MrBayes ver.3.2

Values at nodes represent posterior probabilities >0.5. Phylogenetic clades are represented as in Schrimpf et al. (2014) (Lineages 1-4) and Laggis et al. (2017) (Group 1 and 2), and the position of new haplotypes is indicated by black dot at the end of branch.

Click here for additional data file.

We would like to thank Dr. Martina Jaklič for her help in collecting samples in Slovenia, also we would like to thank Comisia pentru Ocrotirea Monumentelor Naturii for enabling field work in Romania. Our gratitude goes to prof. Christopher M. Austin and Abigail Stancliffe-Vaughan for English language editing and corrections, as well as valuable comments that helped improving the original version of the manuscript. Finally, we would like to give our thanks to the three anonymous reviewers for their constructive criticism.

Additional Information and Declarations

Competing Interests

Author Contributions

Field Study Permissions

Data Availability

The authors declare there are no competing interests.

Riho Gross, Mišel Jelić and Ivana Maguire conceived and designed the experiments, performed the experiments, analysed the data, prepared figures and/or tables, authored or reviewed drafts of the paper, and approved the final draft.

Leona Lovrenčić and Frederic Grandjean performed the experiments, analysed the data, prepared figures and/or tables, authored or reviewed drafts of the paper, and approved the final draft.

Simona Ðuretanović, Vladica Simić, Oksana Burimski and Marius-Ioan Groza performed the experiments, authored or reviewed drafts of the paper, and approved the final draft.

Lena Bonassin analysed the data, prepared figures and/or tables, authored or reviewed drafts of the paper, and approved the final draft.

The following information was supplied relating to field study approvals (i.e., approving body and any reference numbers):

Samples were collected with permissions from local authorities: Croatia –the permission from Ministry of Environmental Protection and Energy of the Republic of Croatia (UP/I-612-07/18-48/148); Serbia - the permissions from Ministry of Environmental Protection, Republic of Serbia Number: 324-04-10/2021-04 and Institute for nature conservation of Serbia, Republic of Serbia 03 No. 026-419/2; and Slovenia –permission from the Environmental Agency of the Republic of Slovenia (35601-1262 150/2006-6 and 35601-135/2010-9), and in Romania samples were collected under permission of Comisia pentru Ocrotirea Monumentelor Naturii Nr. 408/CJ/27.11.2018.

The following information was supplied regarding data availability:

The 16S (MW726211 –MW726336) and COI sequences (MW726338 –MW726635) are available in GenBank.

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
