# Peer review of "Genetic diversity and structure of the noble crayfish populations in the Balkan Peninsula revealed by mitochondrial and microsatellite DNA markers"

_PeerJ, doi:10.7717/peerj.11838_

## Round 0.1 · original submission · Minor Revisions

I agree with reviewers that the manuscript scope brings light to the genetic diversity of crayfish from extent areas of Europe, with clearly stated knowledge gaps and how the manuscript addressed them. I personally liked how the authors formulated the Discussion section. I only have some minor comments in addition to what has been commented on by the three reviewers:

1. kindly add coordinates into the two genetic landscape maps to facilitate an easier understanding of the geographical locations.

2. Also, as mentioned by Reviewer 1, would the small sample size of 9 at Plitvice and Bezid have any effect or sufficient to ensure accuracy of population genetic parameters?

Reviewer 1 ·

Basic reporting

-The article is written in clear, unambiguous, technically correct English text.
A few points that need attention:
L265. Α comma after ‘(655 bp long)’ is needed.
L446. ‘Study also revealed’. The word ‘also’ might be removed.
L779, L913. Journal in italics.
L983-984. ‘Add your references here’ has to be removed. Please check the references’ formatting one more time.

-The article includes sufficient introduction and background information and the study fits well into the broader field of knowledge for the species.

-The article is well structured and all Figures and Tables are relevant to its content with sufficient resolution, appropriately described and labelled. Also, all appropriate raw data have been made available in GenBank.
A few points that need attention:
FIGURE 1. In the down-right corner of the figure, it is stated the Black Sea. This is actually part of Greece and it is the Aegean Sea. The Black Sea is not visible in this figure.
TABLE 1. For site Prespa, the coordinates indicate L. Prespa as the sampling site but it is stated L. Ochrid as the river basin and major tributary. It is not very clear.
L292-297. Are you referring to Fig. 2A, particularly?
L301. Please correct ‘3)’ to ‘(Fig. 3)’.

-The article is self-contained, representing a unit of publication, and includes all results relevant to the hypothesis.

Experimental design

-The article contains clearly defined and meaningful research questions. The knowledge gap being investigated is also stated clearly and statements are made as to how the study contributes to filling that gap.

-The investigation has been conducted rigorously and to a high technical standard. The research has been conducted in conformity with the prevailing ethical standards in the field.
Just a point that needs attention:
The number of samples per population varies from 9 to 33 (Table 1). Nine individuals are quite a few to statistically represent a population. How did you confront this issue? Please mention it also in the Discussion.

-The Materials and Methods of the study are described with sufficient information, in general, to be reproducible by another investigator.
A few points that need attention:
L208. ΦST is presented (also in L345, L346 L556) instead of the FST fixation index. Please explain.
L227-228. Please provide details on how the microsatellite genotypes were scored. Genotyping is a very bias-sensitive procedure. Have they been scored automatically? Were they manually double-checked and from how many experts? Is there any sample, or more, double analysed to ensure the genotyping consistency? What else did you do to avoid any relevant bias?

Validity of the findings

-Decisions are not made based on any subjective determination of impact, degree of advance, novelty or being of interest to only a niche audience. We will also consider studies with null findings. Replication studies will be considered provided the rationale for the replication, and how it adds value to the literature, is clearly described. Please note that studies that are redundant or derivative of existing work will not be considered.

-The vast majority of data on which the conclusions are based is provided or made available in an acceptable discipline-specific repository. The data seems robust, statistically sound, and controlled.
Just a point that needs attention:
L363-364. Why are data not shown? There are other 3 instances in which data are not shown (L350, L367 and L405). Could you please explain each one?

-The conclusions are appropriately stated and connected to the original questions investigated, limited to those supported by the results.

Additional comments

I have read thoroughly the article ‘Genetic diversity and structure of the noble crayfish populations in the Balkan Peninsula revealed by mitochondrial and microsatellite DNA markers’ which address the understudied genetic diversity and population divergence of noble crayfish populations from the western part of the Balkans using mitochondrial and microsatellite DNA markers. The study fills a knowledge gap and provides useful data for conservation planning and management of the species populations.
I would like to congratulate the authors for their clearly written manuscript which is based on large data set, compiled over many years of detailed fieldwork (2008-2016). If I must state some points, as I stated above and in order of importance, which should be improved upon before Acceptance, these are:
1. Clarification on the sample sizes bias of the studied populations
2. Microsatellites scoring
3. ΦST vs FST usage
4. Explanation of why some data are not shown
5. Figure1 and Table 1 corrections
6. Just a few linguistic corrections

Reviewer 2 ·

Basic reporting

no comment

Experimental design

no comment

Validity of the findings

no comment

Additional comments

The finding of this research is very interesting. I would like to suggest the editor accept this manuscript with minor revision. The manuscript is well structured and written, however, there are still minor grammatical mistakes that can be observed throughout the manuscript. The methodology used in this research is satisfactory and suitable to be published in this journal. Even though this project is regional, the data presented are sufficient and discussed well, important as a reference for future research due to the translocation of this species to many places of the world for human consumption. I suggest the authors improve:
- Figure 1: enlarge the country name on the map.
- Figure 2: make all the characters there readable.
- Please label the location name of the black dots in Figure 6.

Reviewer 3 ·

Basic reporting

no comment

Experimental design

no comment

Validity of the findings

no comment

Additional comments

Dear authors,
I was pleased and happy to read through your manuscript bringing light into genetic diversity of native European crayfish species – Astacus astacus. As authors declared, there have been several studies already published, but not considering samples from extant area of southern Europe, especially Balkan countries. That countries have been previously marked as possible genetic diversity refugia for Austropotamobius species and as proven by this study for A. astacus as well. Big effort was done on completing all existing sequences of A. astacus available on GenBank and BOLD databases to present complex structure and diversity of this species. Methods are written clear and detailed. Results are presented precisely with nice figures. Discussion is extensive considering each possible explanation of the results.
I have only very minor comments to the text
Figure 2. The Bayesian phylogram is not well readable due to low resolution. I hope that in final state this will have sufficient resolution. However as this was the case of most of the figures, I guess this is caused due to compression when pdf version of submission was created. Anyway, in the Figure 2, the labels of phylogram are not readable.
Another mistake was found in Supplementary figure 1, where the most of picture is black and all labelling is not visible at all. I don’t know whether this was purpose or a weird artefact, but it needs to be displayed correctly.
Lines 185-186: I would delete bracket from “dataset I) and II)” just dataset I and dataset II are fine.
Lines 423-424: “The lowest Ne values were found in the Romanian population of BEZ (Ne = 11), and the Serbian populations of GRL (Ne = 9) and GAZ (Ne = 18).” – I would switch the two last values being in descending order, i.e., GAZ and GRL.
Lines 495-507: Authors discuss the low genetic diversity of A. astacus comparing to other European crayfish species, however not mentioning the possible relationship to A. balcanicus, the valid species according to Crandall and DeGrave (2017) and their „An updated classification of the freshwater crayfishes (Decapoda: Astacidea)..”. Although, Lake Ohrid is the type locality of this species, adopting sequences from Mrugala et al (2017), the haplotypes were not different enough to show different species status. Most likely also some populations from Laggis et al. (2017) should originated from Vardar river basin, but again didn’t show sufficient divergence. I found the recent paper discussed A. balcanicus status, however dealing with another species A. colchicus (Blaha et al. 2021 Integrative Zoology 2021; 16: 368–378). I would be grateful to authors to add several sentences discussing this issue.

---

## Round 0.2 · Minor Revisions

Dear authors,

I would like to congratulate you on the detailed revision and rebuttal. The current version of the manuscript is almost acceptable. However, I would suggest to the authors to include the description of the sampling sites regarding the geographical connectivity of Prespa Lake and Ohrid Lake (as you have described in the response letter) in the Materials and Method section. This will facilitate an easier understanding of the readers, especially those who are not familiar with the sampling locations.

---

## Round 0.3 · accepted · Accept

Thank you authors for your cooperation and congratulations on the acceptance of your work. It has been a great pleasure editing this manuscript!